# Volume–Outcome Relationship in Cancer Survival Rates: Analysis of a Regional Population-Based Cancer Registry in Japan

**DOI:** 10.3390/healthcare11010016

**Published:** 2022-12-21

**Authors:** Yoichiro Sato, Rena Kaneko, Yuichiro Yano, Kentaro Kamada, Yuui Kishimoto, Takashi Ikehara, Yuzuru Sato, Takahisa Matsuda, Yoshinori Igarashi

**Affiliations:** 1Department of Gastroenterology and Hepatology, Kanto Rosai Hospital, Kanagawa 211-8510, Japan; 2Division of Gastroenterology and Hepatology, Department of Internal Medicine (Omori), School of Medicine, Faculty of Medicine, Toho University, Tokyo 143-8541, Japan; 3Department of Public Health, Graduate School of Medicine, The University of Tokyo, Tokyo 113-0033, Japan

**Keywords:** cancer survival, prognosis, hospital volume, propensity score matching, multiple imputation

## Abstract

Background: There is limited data on the relationship between hospital volumes and outcomes with respect to cancer survival in Japan. The primary objective of this study was to evaluate the effect of hospital volume on cancer survival rate using a population-based cohort database. Methods: Using the Kanagawa cancer registry, propensity score matching was employed to create a dataset for each cancer type by selecting 1:1 matches for cases from high- and other-volume hospitals. The 5-year survival rate was estimated and the hazard ratio (HR) for hospital volume was calculated using a Cox proportional hazard model. Additional analyses were performed limited to cancer patients who underwent surgical operation, chemotherapy, and other treatments in each tumor stage and at the time of diagnosis. Results: The number of cases with complete data, defined as common cancers (prostate, kidney, bladder, esophagus, stomach, liver, pancreas, colon, breast, and lung), was 181,039. Adjusted HR differed significantly among hospital volume categories for the most common cancers except bladder, and the trends varied according to cancer type. The HR ranged from 0.76 (95%CI, 0.74–0.79) for stomach cancer to 0.85 (0.81–0.90) for colon cancer. Conclusions: This study revealed that a relationship may exist between hospital volume and cancer survival in Japan.

## 1. Introduction

For decades, there has been an ongoing discussion concerning the relationship between hospital volume and disease outcome [1]. According to the “volume-outcome” theory, disease prognoses are better at high-volume hospitals than at other-volume hospitals [1].

With respect to health management of cancer in Japan, which is designated as a “super-ageing country” and where cancer has been the major cause of death [2], various policies have been implemented since 1984 as part of the Comprehensive 10-Year Strategy for Cancer Control [3]. In 2001, a designated cancer hospital system was established with the aim of ensuring cancer care, which was augmented in 2006 to cover prefectures and secondary medical areas [4]. In 2006, the Cancer Control Act was approved, and the Basic Plan to Promote Cancer Control Program [5] at the nationwide level was launched in 2007 [6]. The first plan of the Cancer Control Promotion Council covered 5 fiscal years (2007–2011) and was limited to medicine, promotion of chemotherapy and radiation, and training of specialized doctors at designated cancer hospitals. The phase-two plan, beginning in 2012, was broadened to include social undertakings such as training of staff specialized in cancer medical care including cancer specialists [3]. Phase three, initiated in 2018, reinforces phase two and promotes a society living in harmony with cancer.

Patients tend to have different impressions regarding high- and other-volume hospitals with respect to treatment strategies and capabilities. The above measures may make the impression even stronger. High-volume hospitals tend to emphasize “best” outcomes, and patients who are treated there believe that they are receiving the “most professionalized” medical care [7,8]. This is associated with the fact that hospitals with large case volumes are likely to be equipped with greater resources, advanced medical infrastructure, and well-trained staff. In high-volume hospitals, staff experience and clinical pathways improve with the knowledge gained from treating high numbers of similar cases [9]. High-volume hospitals are also able to maintain higher level skills by performing procedures more frequently [10], which is referred to as the “ practice makes perfect” effect [10]. The passion of the physician and staff, independent of the facility environment, may lead to better results. Especially in selected surgical departments, the use of medical care services tends to be dominated by more enthusiastic physicians, a phenomenon referred to as the “enthusiasm hypothesis” [8,11].

Accordingly, diversity among patients’ preferences regarding hospital volume results in differences in the characteristics of hospital cases. In addition, there is potential for healthcare providers to further widen the range of outcomes of treatment. 

Thus, understanding the volume–outcome association in Japan as a consequence of national cancer policies helps to ensure further optimization of the allocation of and benefit from limited healthcare resources not only for patients and healthcare professionals but also hospital administrators and policymakers [12,13]. To date, most volume–outcome studies in Japan have been limited to a single cancer type [14,15,16,17,18], and only a few volume–outcome analyses have been based on population-based cancer registries [12,19]. The assessment of whether differences in healthcare across facilities result in different disease outcomes must be analyzed with strict covariate adjustment [19]. Therefore, in this study, we analyzed the volume–survival relationship using propensity score matching to match baseline characteristics and evaluate whether there is a volume–outcome relationship with respect to cancer survival in Japan.

## 2. Materials and Methods

### 2.1. Data

Kanagawa Prefecture is a neighboring prefecture of Tokyo, and it is the second largest in Japan, with a population of roughly 9 million. The prefecture started its own regional cancer registry in 1970, the fifth in the country, and the accumulated number of cases reached approximately 1,490,000 in 2020. Because the Tokyo Prefecture has only had a registry of cancer cases since 2012 and has, therefore, not yet accumulated substantial data, the Kanagawa Cancer Registry (KCR) is one of the largest regional cancer registries in Japan. Details on the cancer registry system in Japan have been reported elsewhere [20]. Data were collected from neoplasm registration sheets reported by each diagnosing hospital or from clinic and death certificates of residents in Kanagawa Prefecture. In Japan, the National Cancer Registry was initiated under the Law Concerning the Promotion of Cancer Registries, which came into effect in 2016, but Kanagawa Prefecture has maintained the operation of its regional cancer registry, including TNM stage information collected as a separate additional item, given the continuity of data that has been continued to date [21].

The Kanagawa Prefectural Cancer Center collected and consolidated the data into anonymous formats and made these available for academic and administrative purposes.

Accumulated data included the following items: (1) personal identification code, (2) method of registry entry, (3) diagnosing institution, (4) sex, (5) date of birth, (6) date of diagnosis, (7) local government code for the patient’s home address, (8) ICD-10 code for disease name, (9) ICD-O-3 code for pathology, (10) initial or recurrent tumor, (11) therapeutic strategy (very brief), (12) operative procedure (if any), (13) date of death, (14) cause of death, (15) date of last follow-up, and (16) TNM classification and pathological grade according to ICD-O-3 in diagnosed patients. Cancer registries use Union for International Cancer Control (UICC) TNM codes (*UICC TNM classification of malignant tumors*, 6th edition: 2002–2011, 7th edition: 2012–2017, 8th edition: 2018 onwards) to record cancer stages, except for liver cancer, which is staged using the Japanese staging standard (the *General Rules for the Clinical and Pathological Study of Primary Liver Cancer*, 4th edition: 2011, 5th edition: 2012–2018 [22]). To calculate the 5-year survival rate, the prognosis of those with no death information was investigated by batch matching with the Basic Resident Registration Network System and referral of certificate of residence [21]. For example, in 2019, a prognostic study of cases registered in 2013 was carried out, and the study has been continued every year. The proportion of death certificate notification (DCN) cases in the whole database was 5.2% by the end of 2017 [21]. 

### 2.2. Statistical Analysis

With respect to the analysis of cancers, we selected the 10 most common types according to national statistics in Japan [23,24,25], i.e., cancer of the prostate, breast, kidney, bladder, esophagus, stomach, liver, pancreas, colon, and lung. The prevalence rates of these cancers matched almost identically with Japanese national statistics [26]. An initial registry for 1 type of cancer in 1 patient was used. The hospital affiliation of each case was, firstly, the notified hospital adopted, and for cases without a notified hospital, the deceased patient’s hospital. Hospital volume was examined as an independent categorical variable. The number of registered cases between 1990 and 2018 was categorized by organ and sorted by hospital in descending order. The top hospitals that reached 25% of the total number of cases were defined as high-volume hospitals. 

We performed propensity score matching (PSM) since the baseline characteristics were biased with respect to hospital volume. The multivariable logistic regression model for propensity score matching included the following parameters: age at diagnosis, sex, period of diagnosis, and tumor staging. One-to-one matching without replacement was completed using the nearest neighbor match with caliper width set 0.2 times the standard deviation of the logit of the propensity score. Standardized differences were estimated after matching to evaluate the balance of covariates; small absolute values <0.1 SD indicated balance between the groups [27]. A sensitivity analysis to evaluate the impact of an unmeasured confounder was performed using Rosenbaum’s bounds test [28]. The 5-year survival rate was estimated using the Kaplan–Meier method with right censoring at 1825 days. *p*-values were calculated from log-rank tests. Cox proportional hazard model was used to calculate the adjusted hazard ratio (HR) for hospital volume using overall survival (OS) adjusted for basic characteristics (age at diagnosis, gender, calendar period, and tumor staging). Proportional hazard assumption was visually validated using log–log plots. The survival analysis was conducted by dividing the data into 3 categories: (1) PSM overall data; (2) data for different time periods before and after 2006 when the Basic Plan to Promote Cancer Control Programs was enacted in Japan; and (3) to take into account the impact of treatment, an analysis was also conducted restricting the TNM available to those who underwent surgery at stage 1, those who received some form of treatment at stages 2 and 3, and those who received chemotherapy at stage 4. An additional analysis was performed on the whole numbers data for which the tumor staging was supplemented using the multiple imputation (MI) by chained equation method [29]. Five imputed datasets were generated and the HR for hospital volume was calculated.

Chi-square test was performed to analyze differences in baseline characteristics. For continuous variables, Mann–Whitney U test was used to compare groups. The period of diagnosis was categorized into calendar periods for analysis (1990–1995, 1996–2000, 2001–2005, 2006–2010, 2011–2013).

All *p*-values were 2-sided, and *p* < 0.05 was considered statistically significant. All analyses were conducted using Stata/MP15.0 software (Stata Corp LP, College Station, TX, USA). The study was approved by the ethics committees of The University of Tokyo (No. 10891) and Kanto Rosai Hospital (No. 2019-26).

## 3. Results

The total number of patients with cancer in any region registered in the KCR from 1990 to 2018 was 837,164. Within these cases, the details of the raw data that were available are shown as a flow chart in Figure 1. Excluding cases with recurrent enrolment, cases registered since 2014, cases with an invalid birth date or date of registration, and cases of loss to follow-up, 482,256 cases were included among the 10 eligible cancer types. Furthermore, excluding cases from hospitals outside Kanagawa Prefecture and cases without TNM staging, the final number of registered cases falling into the 10 cancer types amounted to 181,039 cases up to 2013.

A total of 837,164 patients (from 1990 to 2018) were identified in the database of the Kanagawa Cancer Registry to reach the final number of eligible patents included in the survival analysis.

The distribution of baseline characteristics of high-volume and other hospital cases is shown in Table 1. The number of hospitals included ranged from 73 to 160 for each cancer. About 3%-9% of high-volume hospitals covered 25% of the caseload. The hospitals that were classified as high-volume fell into the cancer center hospital designation, all but one facility for colon cancer. One facility significantly increased its beds from 224 to 720 in 2000. For bladder and breast cancer, this hospital was newly included as a large hospital after 2005, but for other cancer types there was no change in hospitals that were high-volume hospitals before or after 2005. The average ages of the two groups were similar but did show a statistically significant difference. With respect to sex ratio, esophagus, breast, and lung cancer showed a statistically significant difference. For all cancers, the number of registered cases increased over time. The proportion of registered stages showed similar trends among high-volume and other-volume hospitals, although the trends differed for each cancer. Table 2 shows the distribution of characteristics after propensity score matching. Subsequent to matching, the distribution of characteristics was similar in each cancer category and the balance between the two groups was adjusted. 

The 5-year survival rates for each cancer and the hazard ratios for high-volume hospitals compared to other hospitals are shown in Table 3. Esophagus, stomach, and lung cancer had predominantly low HRs for high-volume hospitals vs other volumes for all eras and at all stages (adjusted HR ranged from 0.71 to 0.85 in esophagus, 0.69 to 0.82 in stomach, and 0.78 to 0.90 in lung cancer). While there was no difference with respect to tumor stage with treatment, prostate, kidney, and breast cancer had a predominantly better prognosis for high-volume hospitals since 2006 (5-year survival of high vs other hospital volumes: 89.9% vs 84.2% in prostate, 74.8% vs 70.0% in kidney, and 88.6% vs 87.3% in breast cancer (adjusted HR: 0.84, 0.94, and 0.85, respectively). For liver and pancreas cancer, the prognosis for high-volume hospitals was significantly longer for stage 1 and stages 2+3. In addition, the prognosis for high-volume hospitals was predominantly prolonged after 2006 (adjusted HR: 0.82 and 0.84, respectively), although there was no difference until 2005. 

Among the 458,965 cases that fell into the 10 target cancers, 267,926 cases lacked tumor staging and were complemented using the MI method. Five imputed datasets for each cancer were made. The combined adjusted HR for high-volume hospitals was significantly lower for almost all cancers except for bladder: prostate, 0.85 (95%CI, 0.78–0.92); kidney, 0.81 (0.71–0.94); bladder, 0.95 (0.84–1.08); esophagus, 0.79 (0.75–0.85); stomach, 0.76 (0.74–0.79); liver, 0.83 (0.78–0.88); pancreas, 0.84 (0.79–0.88); colon, 0.86 (0.82–0.90); breast, 0.84 (0.76–0.92); and lung, 0.84 (0.82–0.87). 

Figure 2 shows Kaplan–Meier survival estimate curves for 5-year overall survival using 181,039 PSM cases with complete information on tumor staging. Log-rank tests revealed a significant difference in prognosis among high-volume and other hospitals for the following cancers (*p* < 0.05): prostate, kidney, esophagus, stomach, liver, pancreas, colon, breast, and lung. Our sensitivity analysis suggested that the high-volume hospital efficacy estimates were only mildly robust (gamma: prostate 1.1, kidney 1.2, bladder 1.05, esophagus 1.15, stomach 1.2, liver 1.20, pancreas 1.2, colon 1.15, breast 1.25, lung 1.25) to the presence of an unmeasured confounder and are likely extant.

Figure 3 shows Kaplan–Meier survival estimate curves for 5-year overall survival among patients who underwent surgery at stage 1, those who received some form of treatment at stages 2 and 3, and those who received chemotherapy at stage 4. Log-rank tests showed statistical significance for all cancers except bladder and prostate cancer in stage 1 with operation. In stages 2 and 3 with some form of treatment, seven cancers not including prostate, kidney, and bladder cancer showed significantly long prognosis in high-volume hospitals. In stage 4 with chemotherapy, only esophagus, stomach, and lung cancer showed a significant difference.

In Figure 2 and Figure 3, the Kaplan–Meier curves for prostate, breast, and pancreatic cancer appear roughly equal. Therefore, considering that these significant differences could be due to the large sample size, we calculated the adjusted hazard ratios for subsamples using the bootstrap method (Appendix A, Table A1). The number of bootstrap iterations was set at 100. Significant differences in HRs for prostate, kidney, and breast cancers disappeared when the sample size was set below 1000, but the advantage was still observed for other cancer types.

## 4. Discussion

Our study yielded the notable finding that there is a meaningful association between hospital volume and survival rate for almost all common cancers in Japan. The adjusted HR differed significantly among high- and other-volume hospitals for almost all common cancers, and the trends varied according to cancer type. The strength of the present report is the large sample size. However, regarding this point of view, caution should be paid to the interpretation of the results because large sample sizes tend to highlight statistical significance even if the effect size is clinically irrelevant.

The most striking result was the better prognosis in high-volume hospitals which was observed for all cancers except bladder cancer since 2006. According to the Basic Plan to Promote Cancer Control Programs [5], designated cancer hospitals promote specialized cancer treatment in terms of human resources and facilities. Not only the treatment but also consultation support or information on cancer care were actively provided by these hospitals, and the quality of life for cancer patients improved and came into live easier [30]. The prevalence of this policy is thought to have created better prognoses at high-volume hospitals. 

In pancreatic cancer, which originally had an extremely poor prognosis, and in liver cancer, where there was no breakthrough chemotherapy for stage 4 before 2013 except for sorafenib [31], high-volume hospitals performed well in stage 1 and stage 2+3 cancers, for which treatment options were available, but there was no significant difference for advanced-stage cancers. The prognosis for esophageal and gastric cancer has improved dramatically due to the possibility of curative endoscopic treatment at an early stage [2]. At the same time, proficiency in chemotherapy and radiotherapy in advanced-stage cancer has improved the prognosis for stage 4 cancers [2] and is probably the reason for the superiority of high-volume hospitals at all stages. In terms of endoscopic treatment, it would seem that many endoscopic procedures such as polypectomy can be performed accurately even in low-volume hospitals, which caused no significant difference in adjusted HR for stage 1 colorectal cancer.

The absence of significant differences in the early stages of prostate and breast cancer is due to the extremely good prognosis. In prostate, bladder, and kidney cancer, the superiority of high-volume hospitals is proven in the overall cases or when MI is used. These results indicate that a larger sample size would result in a significant difference. However, the results of bootstrapping showed that for esophagus, stomach, liver, pancreas, colon, and lung cancers, the prognosis is better in large hospitals, even with smaller samples.

The study suggests that there may be a difference in life expectancy with regard to cancer between high-volume and other-volume hospitals. However, some limitations should be noted in the present study. 

Firstly, a fundamental limitation of this study was that we could not adjust for comorbidities, which have a strong prognostic impact, due to lack of information. Complications such as diabetes and gastric ulcers may have a stronger impact on the prognosis of relatively slowly progressing or early-stage cancer types than the cancer itself or may reduce the effectiveness of cancer treatment [32]. In particular, with respect to the localized stage, the 5-year survival rate for prostate, stomach, colon, and breast cancers is generally greater than 90% [33]. It is possible, therefore, that concomitant disease prior to the start of treatment may have determined the life expectancy, and this may have accounted for the difference in cause of death between the two groups. Furthermore, treatments including postoperative surgical complications (HR, 0.63; 95%CI, 0.50–0.79) and acute medical complications (0.63; 0.48–0.81) have been reported to influence the prognosis [34]. The type of anticancer drugs can also be a strong risk factor for patients’ survival. 

Secondly, we used PSM but this adjusts only for variables included to develop scoring. There is a risk of unmeasured confounding, as comorbidity and hospital type are clearly predicted to be confounding factors. For example, intensive care is preferably provided in teaching hospitals, which tend to exist in urban areas, rather than nonteaching hospitals [8]. The geographical distribution of high-volume teaching hospitals [35,36] may also lead to uneven access to treatment [37]: elderly people with many complications have difficulty in getting to distant hospitals, which can lead to an imbalance of patients. The sensitivity analysis showed that the results were only mildly robust. Therefore, the assumption of unbiased propensity score matching may be false. In the present study, inverse probability weighting (IPW) was not used as the aim was to capture survival with the Kaplan–Meier curve; it should be noted that the results from PSM are the average treatment effect on the treated (ATT), not the average treatment effect (ATE), and therefore do not reflect population effects but only look at effects within matched cases [27].

Moreover, a time break was made in 2005 due to a policy change, but the small number of cases up to 2005 means that there is likely to be bias, regardless of whether PSM was performed or not.

Third, Figure 1 shows that 267,926 cases were excluded because they were not TNM-staged, which may be a major biasing factor. As the Japanese cancer registry only forces large hospitals to determine TNM staging, a study limited to cases that include this variable would inevitably exclude small hospitals. Thus, this could result in a mere hospital-based study, even though it uses data that should have been collected on a population basis. One way we addressed this issue was by using the MI method. However, there is the issue of validity for a complement of nearly 70% of missing values for a single variable. Although there is a lack of evidence in the literature on how estimates derived from MI vary with the amount of missing data, simulation results have reported that complementing 90% of the missing values would give the same result as complementing 1% or 5% of the missing values [38]. According to this theory, this problem may be rectified. 

Fourth, the definition of high vs other volume is a critical problem when assessing the volume–outcome relationship. At present, there are no definitive criteria for assessing hospital volume in Japan [19]. Many similar studies in Japan have used quartiles to define a high volume in terms of the number of registrations by organ and the number of surgeries [17,19,39,40]. This study follows that definition. In Japan, when the period of aggressive cancer treatment comes to an end, patients are often transferred to smaller hospitals for palliative care. However, in rare cases, a patient is confirmed in a small hospital and then transferred to a larger hospital for treatment until deceased. Such cases include prostate 84 (0.3%), kidney 0 (0%), bladder 25 (0.5%), esophagus 39 (0.45%), stomach 163 (0.47%), liver 58 (0.64%), pancreas 115 (1.2%), colon 84 (0.34%), breast 53 (0.19%), and 206 (0.6%) lung cancers. When these cases were changed to the high-volume category and the same analysis was performed, there was no change in the trend of HRs in high-volume hospitals for each cancer. The validity of the definitions used remains an issue for future research, as they may render the results unreliable. 

## 5. Conclusions

We have documented the possible existence of a volume–outcome relationship with respect to the prognosis of common cancers in Japan. The ensuring of cancer care does not lead to uniform outcomes across all hospitals, and it can be said that it has led appropriately to reforming the structure of regional medical care with the aim of prolonging prognosis. This study’s findings can be utilized by physicians, other healthcare personnel, patients, and policymakers regarding clinical and socioeconomic factors that are important in determining the optimal cancer treatment strategy.

## Figures and Tables

**Figure 1 healthcare-11-00016-f001:**
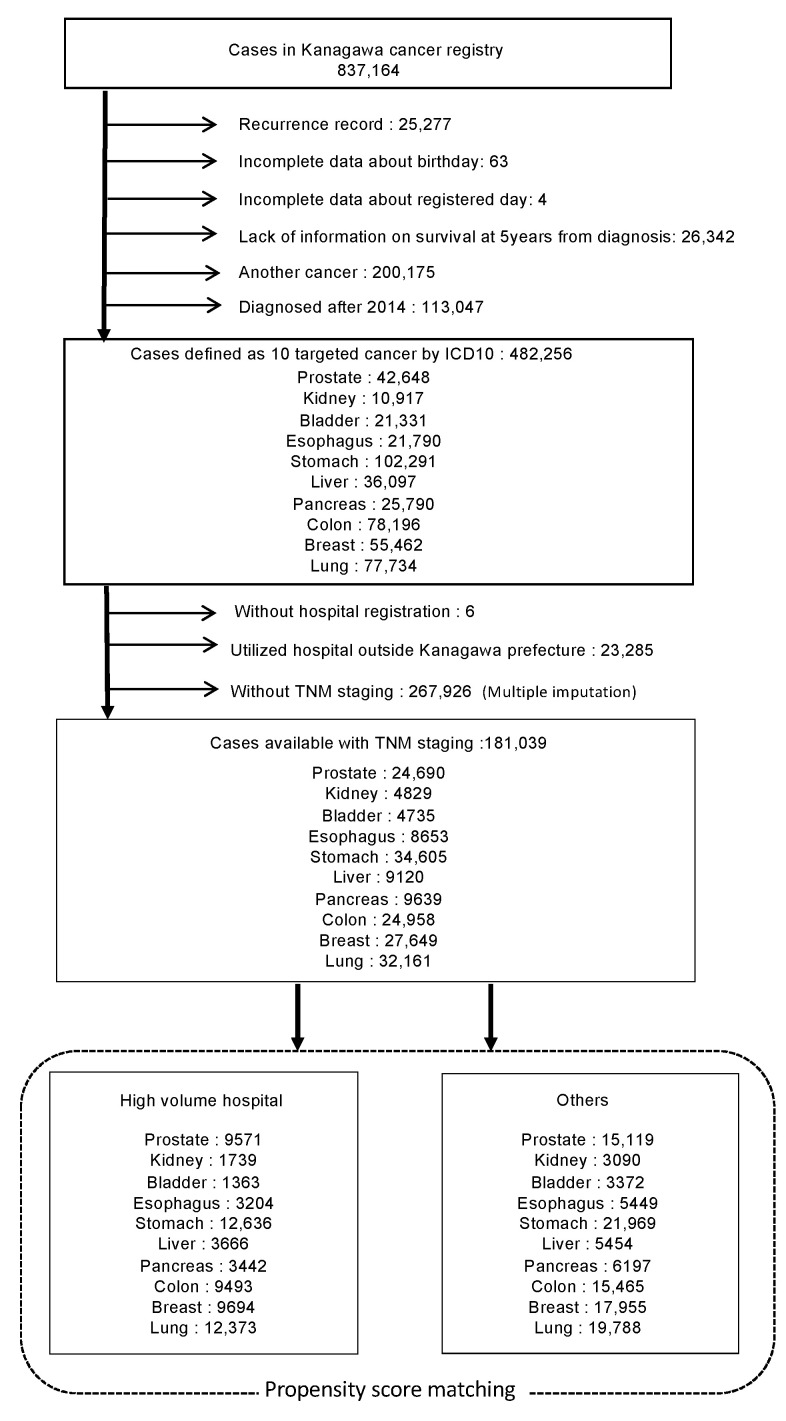
Flow chart of data selection for cases of 10 common cancer types.

**Figure 2 healthcare-11-00016-f002:**
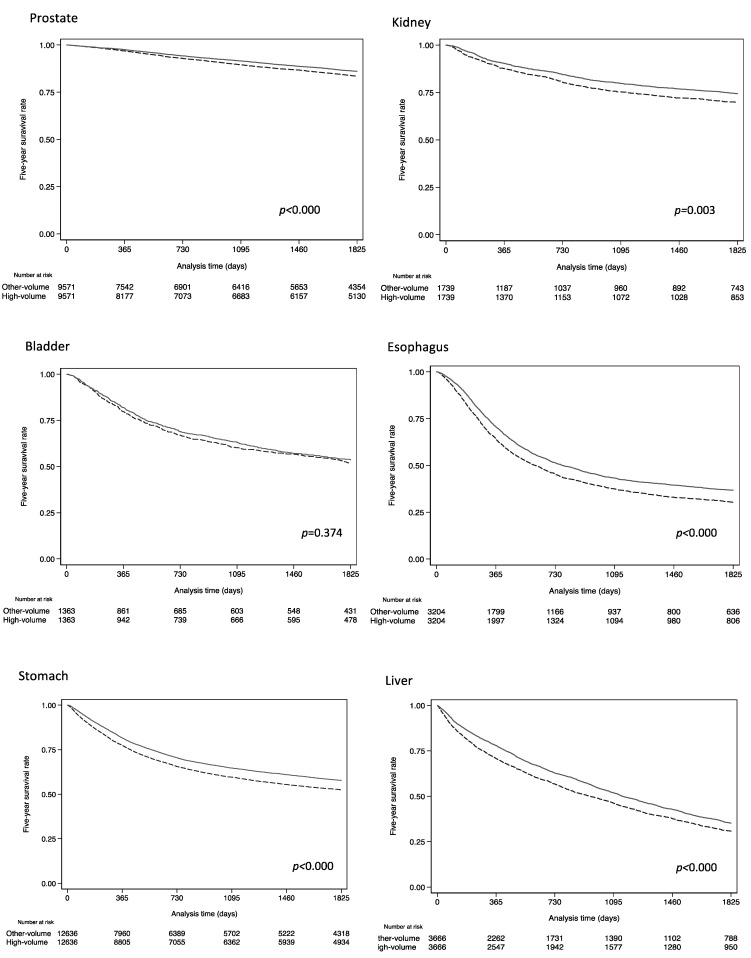
Kaplan–Meier curves for 5-year overall survival in 10 common cancers using propensity score matching. Solid lines indicate the survival for high-volume hospital cases and dashed lines indicate the survival for other hospital cases. The 5-year survival rates for each cancer site at high-volume hospitals was significantly longer except for bladder cancer.

**Figure 3 healthcare-11-00016-f003:**
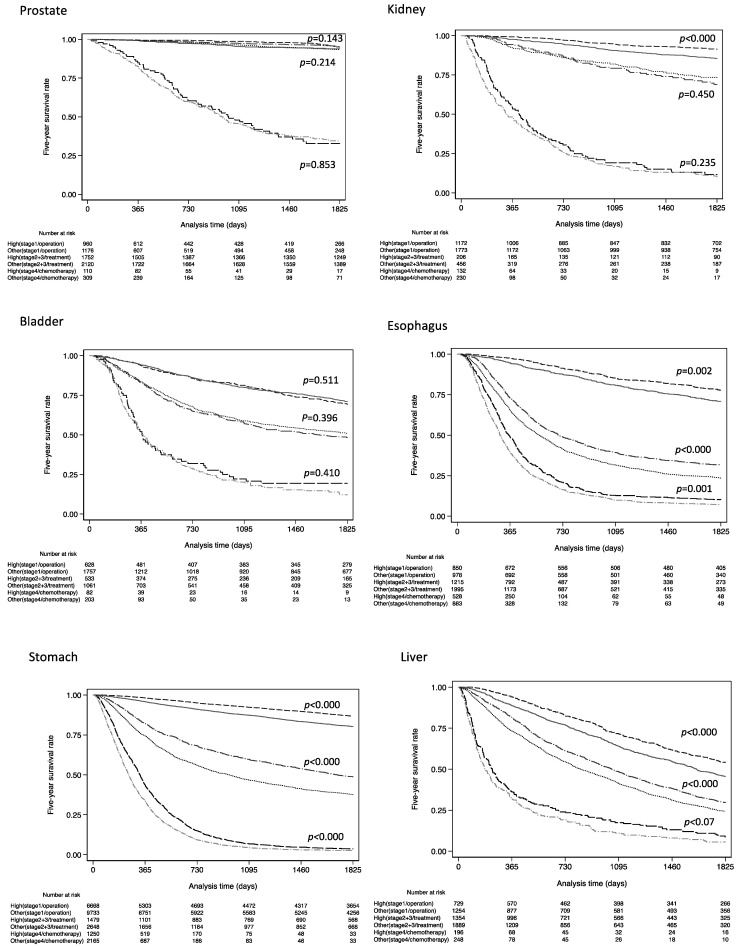
Differences in prognosis by hospital volume for the TNM available to those who underwent surgery at stage 1, those who received some form of treatment at stages 2 and 3, and those who received chemotherapy at stage 4. Log-rank test showed statistical significance (*p* < 0.001) for all cancers except bladder cancer in stage 1 and stages 2 and 3. In stage 4, only esophagus, stomach, and lung cancer showed a significant difference.

**Table 1 healthcare-11-00016-t001:** Baseline characteristics of cases from high-volume and other hospitals.

Cancer Category	High Volume	Others	*p*
Prostate (C619), N	9571	15,119	
The number of hospitals	7	82	
Hospital volume: mean (range) **	129.3 (106.1–155.4)	10.3 (0.6–84.2)	
Gender (male/female), % *	100/0	100/0	-
Age (median:IQR ^§^)	71.2 (66.1:75.9)	73.1 (67.9:78.0)	<0.000 ^‡^
Period of diagnosis (1/2/3/4/5) ^¶^, % *	0.0/0.1/0.7/32.0/67.2	0.0/0.1/1.6/25.4/72.3	<0.000 ^‡^
Stage (1/2/3/4), % *	29.9/45.6/12.2/12.3	29.8/40.5/11.9/17.8	<0.000 ^‡^
Kidney (C649), N	1739	3090	
The number of hospitals	4	69	
Hospital volume: mean (range) **	39.9(32.2–43.4)	2.5(0.6–26.8)	
Gender (male/female), % *	72.3/27.7	72.2/27.8	0.898
Age (median:IQR ^§^)	65.4 (56.5:72.4)	68.5 (60.4:70.6)	<0.000 ^‡^
Period of diagnosis (1/2/3/4/5) ^¶^, % *	0/0.1/1.7/39.3/58.9	0/0.3/1.6/22.9/75.2	<0.000 ^‡^
Stage (1/2/3/4), % *	70.6/5.2/7.5/16.7	64.0/7.0/9.4/9.6	<0.000 ^‡^
Bladder (C670/679), N	1363	3372	
The number of hospitals	7	67	
Hospital volume: mean(range) **	51.9 (45.5–63.3)	2.7 (0.6–40.5)	
Gender (male/female), % *	77.8/22.2	78.7/21.3	0.498
Age (median:IQR^§^)	74.2 (66.2:80.9)	73.8 (66.1:80.3)	0.047 ^†^
Period of diagnosis (1/2/3/4/5) ^¶^, % *	0/0.1/0.5/33/66.4	0.1/0.2/1.7/25.3/72.7	<0.000 ^‡^
Stage (1/2/3/4), % *	47.5/27.2/13.9/11.4	54/21.7/11.9/12.4	<0.000 ^‡^
Esophagus (C150/159), N	3204	5449	
The number of hospitals	3	100	
Hospital volume: mean(range) **	117.2 (72.6–156.1)	3.1 (0.6–47.2)	
Gender (male/female), % *	86.3/13.7	84.6/15.4	0.030 ^†^
Age (median:IQR ^§^)	70.5 (64.1:76.6)	68.4 (63.1:74.3)	<0.000 ^‡^
Period of diagnosis (1/2/3/4/5) ^¶^, % *	0/0.1/5.7/38.0/56.2	0/0.5/4.5/26.8/68.1	<0.000 ^‡^
Stage (1/2/3/4), % *	34.8/13.5/30.7/21.0	27.5/16.1/30.0/26.3	<0.000 ^‡^
Stomach (C160/169), N	12,636	21,969	
The number of hospitals	7	153	
Hospital volume: mean(range) **	222.6 (168.5–331.3)	8.2 (0.6–112.6)	
Gender (male/female), % *	70.6/29.4	70.9/29.1	0.492
Age (median:IQR ^§^)	70.8 (63.8:77.1)	72.7 (65.4:79.1)	<0.000 ^‡^
Period of diagnosis (1/2/3/4/5) ^¶^, % *	0.0/1.0/7.6/35.7/55.7	0.0/0.5/4.3/26.8/68.4	<0.000 ^‡^
Stage (1/2/3/4), % *	58.4/10.1/11.8/19.7	51.5/12.1/10.8/25.6	<0.000 ^‡^
Liver (C220), N	3666	5454	
The number of hospitals	8	86	
Hospital volume: mean(range) **	65.5 (49.0–78.5)	3.2 (0.6–34.1)	
Gender (male/female), % *	70.8/29.1	70.1/29.9	0.485
Age (median:IQR ^§^)	71.5 (64.4:77.4)	73.3 (65.8:79.3)	<0.000 ^‡^
Period of diagnosis (1/2/3/4/5) ^¶^, % *	0.2/0.8/5.7/35.7/57.6	0.0/0.3/2.2/29.3/68.2	<0.000 ^‡^
Stage (1/2/3/4), % *	40.5/27.8/22.2/9.5	37.4/27.1/22.4/13.1	<0.000 ^‡^
Pancreas (C250/259), N	3442	6197	
The number of hospitals	6	103	
Hospital volume: mean(range) **	57.8 (40.7–93.2)	3.4 (0.6–30.9)	
Gender (male/female), % *	57.0/43.0	55.5/44.5	0.166
Age (median:IQR ^§^)	69.6 (63.0:75.8)	73.4 (66.1:80.5)	<0.000 ^‡^
Period of diagnosis (1/2/3/4/5) ^¶^, % *	0.0/0.0/4.2/35.0/60.7	0/0.3/2.9/23.1/73.7	<0.000 ^‡^
Stage (1/2/3/4), % *	5.8/20.1/21.3/52.8	6.8/15.6/15.4/62.1	<0.000 ^‡^
Colon (C180/189), N	9493	15,465	
The number of hospitals	8	123	
Hospital volume: mean(range) **	163.5 (124.8–271.8)	7.1 (0.6–109.7)	
Gender (male/female), % *	54.6/45.4	55.3/44.7	0.257
Age (median:IQR ^§^)	70.9 (63.1:77.7)	72.6 (65.1:79.6)	<0.000 ^‡^
Period of diagnosis (1/2/3/4/5) ^¶^, % *	0.2/1.5/6.4/36.7/55.3	0.0/0.4/4.5/24.7/70.4	<0.000 ^‡^
Stage (1/2/3/4), % *	30.0/19.4/31.1/19.3	26.5/21.5/27.9/24.1	<0.000 ^‡^
Breast (C500/509), N	9694	17,955	
The number of hospitals	4	106	
Hospital volume: mean(range) *	262.4 (168.5–324.1)	9.8 (0.6–136.8)	
Gender (male/female), % *	0.3/99.7	0.6/99.4	<0.000 ^‡^
Age (median:IQR ^§^)	57.3(47.1:67.1)	60.9(50.0:71.6)	<0.000 ^‡^
Period of diagnosis (1/2/3/4/5) ^¶^, % *	0.0/0.5/5.5/34.9/59.1	0.1/1.3/5.6/26.8/66.3	<0.000 ^‡^
Stage (1/2/3/4), % *	44.9/38.9/10.6/5.6	41.6/40.3/11.1/7.0	<0.000 ^‡^
Lung (C340/349), N	12,373	19,788	
The number of hospitals	6	125	
Hospital volume: mean(range) **	217.4 (166.5–375.7)	8.8 (0.6–121.9)	
Gender (male/female), % *	68.0/32.0	70.4/29.6	<0.000 ^‡^
Age (median:IQR ^§^)	70.5 (63.6:76.5)	72.7 (65.8:79.1)	<0.000 ^‡^
Period of diagnosis (1/2/3/4/5) ^¶^, % *	0.0/0.5/6.7/36.4/56.4	0.0/0.5/4.8/25.0/69.7	<0.000 ^‡^
Stage (1/2/3/4), % *	37.4/8.7/21.4/32.5	25.8/6.6/21.5/46.1	<0.000 ^‡^

*p*-values <0.01 ^‡^ or <0.05 ^†^ were considered to be statistically significant. * Percentage may not total 100 because of rounding. ** The average of annual number of registered patients per hospital (1996–2013). ^§^ IQR: interquartile range. ^¶^ Period of diagnosis: the period of diagnosis was categorized into 1 (1990–1995), 2 (1996–2000), 3 (2001–2005), 4 (2006–2010), 5 (2011–2013).

**Table 2 healthcare-11-00016-t002:** Baseline characteristics of cases from high-volume centers and matched cases from other hospitals.

Cancer Category	High Volume	Others	SD
Prostate (C619), N	9571	9571	
Gender (male/female), % *	100/0	100/0	N.A
Age (median:IQR ^§^)	71.2 (66.1:75.9)	71.3 (66.2:76.2)	−0.006
Period of diagnosis (1/2/3/4/5) ^¶^, % *	0.0/0.1/0.7/32.0/67.2	0.0/0.2/2.1/29.2/68.6	0.002
Stage (1/2/3/4), % *	29.9/45.6/12.2/12.3	32.9/43.2/10.2/13.8	0.022
Kidney (C649), N	1739	1739	
Gender (male/female), % *	72.3/27.7	72.2/27.8	−0.003
Age (median:IQR^§^)	65.4 (56.5:72.4)	64.8 (55.7:72.7)	0.020
Period of diagnosis (1/2/3/4/5) ^¶^, % *	0/0.1/1.7/39.3/58.9	0/0.5/2.9/34.0/62.6	−0.030
Stage (1/2/3/4), % *	70.6/5.2/7.5/16.7	67.9/7.9/8.3/15.8	−0.005
Bladder (C670/679), N	1363	1363	
Gender (male/female), % *	77.8/22.2	78.0/22.2	0.005
Age (median:IQR ^§^)	74.2 (66.2:80.9)	74.5 (66.8:80.4)	−0.015
Period of diagnosis (1/2/3/4/5) ^¶^, % *	0/0.1/0.5/33/66.4	0/0/2.1/29.1/68.8	−0.019
Stage (1/2/3/4), % *	47.5/27.2/13.9/11.4	49.8/23.2/12.9/14.1	−0.017
Esophagus (C150/159), N	3204	3204	
Gender (male/female), % *	86.3/13.7	86.5/13.5	0.005
Age (median:IQR ^§^)	70.5 (64.1:76.6)	68.6 (62.5:74.8)	0.014
Period of diagnosis (1/2/3/4/5) ^¶^, % *	0/0.1/5.7/38.0/56.2	0/0.9/6.7/33.1/59.3	−0.006
Stage (1/2/3/4), % *	34.8/13.5/30.7/21.0	33.4/17.2/27.6/21.8	0.000
Stomach (C160/169), N	12,636	12,636	
Gender (male/female), % *	70.6/29.4	70.9/29.1	0.014
Age (median:IQR ^§^)	70.8 (63.8:77.1)	70.8 (63.5:77.2)	0.011
Period of diagnosis (1/2/3/4/5) ^¶^, % *	0.0/1.0/7.6/35.7/55.7	0.1/0.9/7.0/34.7/57.4	−0.037
Stage (1/2/3/4), % *	58.4/10.1/11.8/19.7	59.3/11.8/9.5/19.4	0.032
Liver (C220), N	3666	3666	
Gender (male/female), % *	70.8/29.1	70.5/29.5	−0.008
Age (median:IQR ^§^)	71.5 (64.4:77.4)	71.6 (63.9:77.7)	−0.010
Period of diagnosis (1/2/3/4/5) ^¶^, % *	0.2/0.8/5.7/35.7/57.6	0.0/0.4/3.3/38.3/58.0	−0.062
Stage (1/2/3/4), % *	40.5/27.8/22.2/9.5	41.5/28.8/19.4/10.4	0.020
Pancreas (C250/259), N	3442	3442	
Gender (male/female), % *	57.0/43.0	57.8/42.2	0.018
Age (median:IQR ^§^)	69.6 (63.0:75.8)	69.0 (62.6:75.8)	0.012
Period of diagnosis (1/2/3/4/5) ^¶^, % *	0.0/0.0/4.2/35.0/60.7	0/0.4/4.9/30.7/63.9	−0.030
Stage (1/2/3/4), % *	5.8/20.1/21.3/52.8	8.9/18.3/17.0/55.8	0.013
Colon (C180/189), N	9493	9493	
Gender (male/female), % *	54.6/45.4	52.2/44.8	0.012
Age (median:IQR ^§^)	70.9 (63.1:77.7)	70.9 (63.3:77.8)	−0.006
Period of diagnosis (1/2/3/4/5) ^¶^, % *	0.2/1.5/6.4/36.7/55.3	0.0/0.6/7.3/35.7/56.4	−0.035
Stage (1/2/3/4), % *	30.0/19.4/31.1/19.3	29.0/22.7/27.6/20.7	−0.006
Breast (C500/509), N	9694	9694	
Gender (male/female), % *	0.3/99.7	0.3/99.7	−0.004
Age (median:IQR ^§^)	57.3 (47.1:67.1)	57.1 (47.4:66.9)	−0.001
Period of diagnosis (1/2/3/4/5) ^¶^, % *	0.0/0.5/5.5/34.9/59.1	0.2/1.6/6.7/28.2/63.3	−0.002
Stage (1/2/3/4), % *	44.9/38.9/10.6/5.6	44.8/39.8/9.6/5.7	0.006
Lung (C340/349), N	12,373	12,373	
Gender (male/female), % *	68.0/32.0	68.9/31.1	0.020
Age (median:IQR ^§^)	70.5 (63.6:76.5)	70.5 (63.7:76.7)	−0.014
Period of diagnosis (1/2/3/4/5) ^¶^, % *	0.0/0.5/6.7/36.4/56.4	0.1/0.8/7.5/30.6/61.0	−0.048
Stage (1/2/3/4), % *	37.4/8.7/21.4/32.5	0.0/0.5/6.7/36.4/56.4	0.000

* Percentage may not total 100 because of rounding. SD: standard difference. ^§^ IQR: interquartile range. ^¶^ Period of diagnosis: the period of diagnosis was categorized into 1 (1990–1995), 2 (1996–2000), 3 (2001–2005), 4 (2006–2010), 5 (2011–2013).

**Table 3 healthcare-11-00016-t003:** Hazard ratio of high-volume hospitals against other volumes.

	5 Year Survival Rate (%)		Hazard Ratio			
Cancer Category *	High Volume	Other Volumes	Crude HR	95%CI	Adjusted HR **	95%CI
Prostate (C619)						
PSM						
All	87.2	83.7	0.83	0.76–0.90	0.84	0.77–0.91
1990–2005	86.0	83.4	0.63	0.29–1.36	0.71	0.33–1.54
2006–2013	89.9	84.2	0.83	0.76–0.90	0.84	0.77–0.91
Stage						
Stage 1/operation	95.1	94.0	0.66	0.38–1.15	0.77	0.46–1.35
Stages 2+3/treatment	95.4	93.4	0.70	0.52–0.95	0.98	0.72–1.34
Stage 4/chemotherapy	32.7	34.0	0.97	0.73–1.29	1.00	0.75–1.34
Multiple imputation	-	-	0.84	0.76–0.91	0.85	0.78–0.92
Kidney (C649)						
PSM						
All	74.4	69.9	0.81	0.71–0.94	0.79	0.69–0.92
1990–2005	58.1	58.6	1.06	0.54–2.09	1.12	0.46–2.73
2006–2013	74.8	70.0	0.81	0.70–0.88	0.94	0.66–0.89
Stage						
Stage 1/operation	91.4	85.5	0.56	0.43–0.74	0.81	0.68–0.95
Stages 2+3/treatment	69.9	73.4	1.14	0.81–1.62	0.81–1.62	0.87–1.76
Stage 4/chemotherapy	11.8	10.1	0.86	0.67–1.10	0.91	0.72–1.17
Multiple imputation	-	-	0.79	0.69–0.91	0.81	0.71–0.94
Bladder (C670/679)						
PSM						
All	53.6	51.9	0.94	0.83–1.06	0.95	0.85–1.08
1990–2005	53.5	51.1	2.08	0.52–8.34	0.61	0.09–3.87
2006–2013	62.5	79.3	0.92	0.82–1.04	0.95	0.84–1.08
Stage						
Stage 1/operation	69.2	71.0	1.06	0.88–1.29	1.00	0.83–1.22
Stages 2+3/treatment	48.4	50.1	1.07	0.91–1.26	1.04	0.89–1.23
Stage 4/chemotherapy	12.4	12.2	0.88	0.66–1.19	0.89	0.67–1.20
Multiple imputation	-	-	0.94	0.83–1.06	0.95	0.84–1.08
Esophagus (C150/159)						
PSM						
All	36.8	30.1	0.82	0.77–0.87	0.79	0.74–0.85
1990–2005	36.5	25.5	0.63	0.50–0.81	0.71	0.56–0.90
2006–2013	40.8	30.7	0.83	0.78–0.89	0.81	0.75–0.86
Stage						
Stage 1/operation	77.5	70.8	0.71	0.57–0.88	0.75	0.59–0.94
Stages 2+3/treatment	31.8	23.5	0.78	0.71–0.85	0.83	0.76–0.91
Stage 4/chemotherapy	10.2	7.1	0.83	0.74–0.93	0.85	0.75–0.95
Multiple imputation	-	-	0.82	0.77–0.88	0.79	0.75–0.85
Stomach (C160/169)						
PSM						
All	57.8	52.4	0.83	0.79–0.86	0.76	0.74–0.79
1990–2005	57.7	50.7	0.76	0.67–0.86	0.82	0.72–0.93
2006–2013	59.1	52.6	0.84	0.80–0.88	0.76	0.73–0.79
Stage						
Stage 1/operation	86.8	80.2	0.62	0.57–0.69	0.69	0.64–0.77
Stages 2+3/treatment	48.0	37.5	0.71	0.64–0.78	0.80	0.73–0.88
Stage 4/chemotherapy	36.4	24.8	0.78	0.72–0.84	0.78	0.72–0.84
Multiple imputation	-	-	0.83	0.79–0.86	0.76	0.74–0.79
Liver (C220)						
PSM						
All	35.2	30.7	0.85	0.79–0.89	0.83	0.78–0.88
1990–2005	33.9	29.5	1.17	0.86–1.61	1.16	0.84–1.60
2006–2013	57.0	50.6	0.85	0.79–0.90	0.82	0.77–0.87
Stage						
Stage 1/operation	54.2	45.6	0.76	0.66–0.88	0.80	0.69–0.93
Stages 2+3/treatment	29.7	24.6	0.82	0.76–0.90	0.90	0.83–0.98
Stage 4/chemotherapy	8.7	5.6	0.83	0.86–1.02	1.05	0.86–1.29
Multiple imputation	-	-	0.84	0.81–0.90	0.83	0.78–0.88
Pancreas (C250/259)						
PSM						
All	6.3	5.5	0.86	0.81–0.90	0.81–0.90	0.81–0.90
1990–2005	2.0	2.3	1.02	0.81–1.28	0.90	0.71–1.13
2006–2013	6.3	5.2	0.84	0.81–0.89	0.84	0.79–0.88
Stage						
Stage 1/operation	5.0	3.0	0.58	0.44–0.79	0.67	0.48–0.89
Stages 2+3/treatment	0.96	0.84	0.82	0.75–0.88	0.84	0.78–0.91
Stage 4/chemotherapy	0.03	0.03	0.93	0.86–1.00	0.95	0.89–1.03
Multiple imputation	-	-	0.85	0.82–0.89	0.84	0.79–0.88
Colon (C180/189)						
PSM						
All	63.6	58.8	0.86	0.82–0.92	0.85	0.81–0.90
1990–2005	62.1	58.5	0.82	0.69–0.97	0.88	0.84–1.05
2006–2013	64.6	62.1	0.87	0.83–0.91	0.85	0.81–0.89
Stage						
Stage 1/operation	84.8	81.5	0.78	0.68–0.91	0.89	0.77–1.02
Stages 2+3/treatment	70.3	65.8	0.75	0.69–0.81	0.80	0.74–0.88
Stage 4/chemotherapy	14.3	14.6	0.94	0.86–1.03	0.96	0.88–1.05
Multiple imputation	-	-	0.87	0.83–0.91	0.86	0.82–0.90
Breast (C500/509)						
PSM						
All	88.2	86.5	0.83	0.76–0.92	0.83	0.76–0.92
1990–2005	84.9	86.4	1.18	0.89–1.56	1.21	0.90–1.63
2006–2013	88.6	87.3	0.81	0.73–0.89	0.85	0.74–0.89
Stage						
Stage 1/operation	96.7	95.1	0.65	0.51–0.82	0.84	0.66–1.06
Stages 2+3/treatment	88.9	84.4	0.68	0.61–0.77	0.79	0.71–0.90
Stage 4/chemotherapy	31.5	29.5	0.86	0.73–1.02	0.87	0.73–1.03
Multiple imputation	-	-	0.83	0.76–0.92	0.84	0.76–0.92
Lung (C340/349)						
PSM						
All	33.9	28.6	0.85	0.82–0.87	0.84	0.81–0.87
1990–2005	28.6	21.2	0.79	0.72–0.88	0.90	0.82–0.99
2006–2013	34.5	29.5	0.85	0.82–0.88	0.83	0.81–0.86
Stage						
Stage 1/operation	77.5	70.2	0.72	0.65–0.79	0.78	0.70–0.87
Stages 2+3/treatment	28.5	20.4	0.77	0.72–0.81	0.80	0.75–0.85
Stage 4/chemotherapy	5.5	4.8	0.87	0.83–0.92	0.89	0.85–0.95
Multiple imputation	-	-	0.84	0.81–0.87	0.84	0.82–0.87

* Each cancer category was analyzed by PSM: analyzed with propensity-score-matched dataset, and each diagnosis year, by stage: analyzed limited to stage 1 with operation, stages 2+3 with some kind of treatment, and stage 4 with chemotherapy. For analyses using multiple interpolation methods, only HRs are indicated due to the unreliability of the survival estimates. ** Adjusted HR: hazard ratio controlled for sex, age at diagnosis, tumor stage, period of diagnosis. CI: confidence interval.

## Data Availability

The data presented in this study are available on request from the corresponding author. The data are not publicly available due to privacy or ethical restrictions.

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
