# Peer review of "Volume–Outcome Relationship in Cancer Survival Rates: Analysis of a Regional Population-Based Cancer Registry in Japan"

_healthcare, 2022, doi:10.3390/healthcare11010016_

Round 1
Reviewer 1 Report
The authors face a ground-breaking issue trying to provide evidence for a common but still undemonstrated belief that healthcare and clinical outcomes (in particular cancer survival) are strictly related with the hospital cancer-specific case volume. This investigation is paramount since it is related to the several aspects regarding the so-called precision-medicine in which the hospitals and the medical centres are asked to be even more skilled on specific disease. Another strength point of the manuscript is of course the large sample size. However, under this point of view, caution should be paid in the interpretation of results, because the large sample size tends to highlight statistical significance even if the effect size is clinically irrelevant. The manuscript is well written, the aim well defined and the investigation hypotheses are remarkable. However, some methodological weaknesses should be addressed in order to make the manuscript fully worthy for publication. I strongly encourage the Authors to make last efforts for improving the manuscript. Please, see some major and minor points listed below.MATERIAL AND METHODS STATISTICAL ANALYSIS -The hospital affiliation definition (lines 114-116 page 3) could hide a bias. For example, it could happen that unresponsive patients are transferred in a more skilled (and thus with highest volume for the specific cancer) hospital, and, as a consequence, these patients could die in a hospital different with respect to the hospital where they received the first treatments. The Authors should identify these cases (if they exist) and manage them to prevent this potential bias. -The definition of “high volume hospital” in lines 116-119 appears biased. This definition is crucial but the Authors’ choice appears arbitrary, making the findings of the manuscript potentially unreliable. Consider, in fact, the following example with respect to the registered cases for a specific organ: the first 4 hospitals having respectively [15,6,2,2] cases and then the remaining hospitals having 1 case each. In this example, following the high volume definition, the hospitals with 2 cases are defined as high-volume while the hospital with 1 case as low (other) volume hospital. But it is easy to object that from both clinical and administrative point of view, a volume of 1 or 2 cases is substantially the same. The cut-off for discriminating between high and low (or other) volume should be based on clinical or administrative considerations or on official health-care policies. In many European countries, for example, high volume hospitals are defined a priori as at least 50 cases per organ per year. Considering this definition as paramount for the evaluation of all findings, the Authors should provide evidence that their choice is robust or take into consideration to change the definition accordingly with clinical and/or administrative purposes. -About imputation of data (line 138 page 3). Data imputation always introduces a bias (imputed data are estimated with an uncertainty). Considering the large amount of available data, perhaps the Authors might remove the missing data after assessing their ‘missing at random’ nature, without resort to imputation. RESULTS -Figure 1. Please clearly explain on which amount of data (181039 or 267926) the multiple imputation was performed. -Page 5, line 164-165. This significance is likely due to the large sample size. This point should be considered across all manuscript results. -Tables and Figures. The pvalues of ‘0.000’ have no sense. Please substitute with e.g. ‘<0.001’. -Table 2. Although the application of propensity score matching has substantially mitigated the differences between high and other volume hospitals in terms of main investigated features, a clear evaluation of the differences between the two groups, especially for the stage, should be assessed by a statistical test other than by using the SD. See e.g. the stage distribution for Lung: in the not-high volume group the percentage of stage IV appears significantly higher than the corresponding percentage in the High volume group. And this could be related to the better survival outcome for high volume group for lung cancer. In order to vanish any doubts, please add in table 2 the test for assessing the feature difference after groups matching. -Page 8, lines 188-189. This sentence appears to have no correspondence with the results reported in Table 3. With respect to 5-years survival rate, the better prognosis holds for: prostate , kidney and breast cancers, with 5-years survival rates (all cases) range from 74.4, 69.9 for kidney to 88.2 and 86.5 for breast. -Figure 2 and Figure 3. The significance should be considered with caution. Looking e.g. at the prostate panel of figure 2, it appears clear that the two KM curves are substantially equal. The very slight difference reaches the significance only due to the large sample size but clinically (I am afraid) there is no evidence for differences. Similarly, it happens also for Pancreas and Breast. The coherence between clinical vs statistical significance should be taken into account when each finding is commented and discussed. A possible solution for assessing the robustness of results could be, for example, to use a bootstrap procedure on a subsample (smaller sample) of data preventing the pvalue inflations. DISCUSSION I have appreciated the listed limitations of the study, however, other points should be cited coherently with the points/weaknesses highlighted above if these cannot be adequately addressed by the Authors.
Author Response
December 9, 2022
Dear Reviewer
Thank you for your review regarding our manuscript, “Volume-Outcome Relationship in Cancer Survival Rates: Analysis of a Regional Population-Based Cancer Registry in Japan ” which was assigned the manuscript number healthcare-2033848.
I have attached our revised manuscript and provided point-by-point responses to the reviewer’s valuable comments.
We wish to express our appreciation to the reviewer for their comments, which have helped us to significantly improve the paper.
MATERIAL AND METHODS STATISTICAL ANALYSIS -The hospital affiliation definition (lines 114-116 page 3) could hide a bias. For example, it could happen that unresponsive patients are transferred in a more skilled (and thus with highest volume for the specific cancer) hospital, and, as a consequence, these patients could die in a hospital different with respect to the hospital where they received the first treatments. The Authors should identify these cases (if they exist) and manage them to prevent this potential bias.
Response: The number of cases as indicated was calculated and same analysis was performed. The results did not change and this point has been added to lines 338-345 of the discussion.
-The definition of “high volume hospital” in lines 116-119 appears biased. This definition is crucial but the Authors’ choice appears arbitrary, making the findings of the manuscript potentially unreliable. Consider, in fact, the following example with respect to the registered cases for a specific organ: the first 4 hospitals having respectively [15,6,2,2] cases and then the remaining hospitals having 1 case each. In this example, following the high volume definition, the hospitals with 2 cases are defined as high-volume while the hospital with 1 case as low (other) volume hospital. But it is easy to object that from both clinical and administrative point of view, a volume of 1 or 2 cases is substantially the same. The cut-off for discriminating between high and low (or other) volume should be based on clinical or administrative considerations or on official health-care policies. In many European countries, for example, high volume hospitals are defined a priori as at least 50 cases per organ per year. Considering this definition as paramount for the evaluation of all findings, the Authors should provide evidence that their choice is robust or take into consideration to change the definition accordingly with clinical and/or administrative purposes.
Response: As pointed out, the hospital groupings are quite arbitrary. However, there is no definition of a high-volume hospital that is conventionally used in Japan. As most Japanese volume-outcome papers to date have defined high-volume using quartiles of case volumes, we have followed this division in this study. Additions have been made to the limitations at line 335-338 in this regard.
-About imputation of data (line 138 page 3). Data imputation always introduces a bias (imputed data are estimated with an uncertainty). Considering the large amount of available data, perhaps the Authors might remove the missing data after assessing their ‘missing at random’ nature, without resort to imputation.
Response: We are aware of your opinion. However, as the MI results are only a complementary part of this study, please allow us to leave them in as they are at this time. We do not have enough time to change this approach before the deadline. However, we will consider how to do this in the future. Sorry.
RESULTS -Figure 1. Please clearly explain on which amount of data (181039 or 267926) the multiple imputation was performed.
Response: We have changed our notation.
-Page 5, line 164-165. This significance is likely due to the large sample size. This point should be considered across all manuscript results.
Response: The possibility of a difference in dominance by sample size was cautioned (lines 267-270).
-Tables and Figures. The pvalues of ‘0.000’ have no sense. Please substitute with e.g. ‘<0.001’.
Response: The “0.000” notation in the figures and tables has been revised.
-Table 2. Although the application of propensity score matching has substantially mitigated the differences between high and other volume hospitals in terms of main investigated features, a clear evaluation of the differences between the two groups, especially for the stage, should be assessed by a statistical test other than by using the SD. See e.g. the stage distribution for Lung: in the not-high volume group the percentage of stage IV appears significantly higher than the corresponding percentage in the High volume group. And this could be related to the better survival outcome for high volume group for lung cancer. In order to vanish any doubts, please add in table 2 the test for assessing the feature difference after groups matching.
Response: In propensity score matching, the standard differences are unaffected by sample size. A standardized difference <0.1 indicates a negligible difference in the mean of proportion of a covariate between the case and control groups (Yasunaga H. Annals of Clinical Epidemiology 202;2(2)3:33-37, Park Y, et al. BMJ 2018 Mar 28:360:k1218). The literature also states, “showing p values is not recommended to check the balance of covariates, because failure to reject the null hypothesis does not guarantee successful balance of covariates between the groups”. Thus, we showed SD.
A note was added to the method on the points that indicated SD (lines 124-126). In fact, the tested P-values are significantly different for all cancers for both periods of diagnosis and stage. Please let me know if I still need to add more, or if my argument is incorrect.
-Page 8, lines 188-189. This sentence appears to have no correspondence with the results reported in Table 3. With respect to 5-years survival rate, the better prognosis holds for: prostate , kidney and breast cancers, with 5-years survival rates (all cases) range from 74.4, 69.9 for kidney to 88.2 and 86.5 for breast.
Response: Sorry, lines 188 to 189 are correct as an explanation of the results in Table 3. Could you please elaborate on what is wrong? i.e., :“Esophagus, stomach, and lung cancer had a better prognosis for 5-year survival, and predominantly low HRs for high-volume hospital vs other volume for all eras and at all stages (adjusted HR ranged from 0.71-0.85 in esophagus, 0.69-0.82 in stomach and 0.78-0.90 in lung“. ( Now, lines 195-198).
-Figure 2 and Figure 3. The significance should be considered with caution. Looking e.g. at the prostate panel of figure 2, it appears clear that the two KM curves are substantially equal. The very slight difference reaches the significance only due to the large sample size but clinically (I am afraid) there is no evidence for differences. Similarly, it happens also for Pancreas and Breast. The coherence between clinical vs statistical significance should be taken into account when each finding is commented and discussed. A possible solution for assessing the robustness of results could be, for example, to use a bootstrap procedure on a subsample (smaller sample) of data preventing the pvalue inflations.
Response: As indicated, results for different sample numbers using the bootstrap method have been added to the appendix. When the sample size is reduced, significant differences disappear for prostate, renal and breast cancer, so this has been added to the text (lines 248-254, 293-295).
DISCUSSION I have appreciated the listed limitations of the study, however, other points should be cited coherently with the points/weaknesses highlighted above if these cannot be adequately addressed by the Authors.
Response: The issue of significant differences due to sample size, and other issues as pointed out, have been added to the limitations.
We feel that the revised manuscript reflects a suitable response to the comments and has been significantly improved over the initial submission. We trust that it is now suitable for publication in the Healthcare.
Sincerely.
Rena Kaneko M.D, Ph.D
Department of Gastroenterology, Japan Organization of Occupational Health and Safety Kanto Rosai Hospital.
1-1Kizukisumiyoshi-cho, Nakahara-ku, Kawasaki, Kanagawa, 211-8510 Japan
Tel: +81-44-411-3131; Fax: +81-44-411-3650
Reviewer 2 Report
This paper discusses the association between hospital volume and prognosis for major cancers. However, the methods of analysis and the way conclusions are drawn are questionable.
Major point
1)The authors define "High volume hospitals," but were these hospitals never changed in ranking over time?
2) As the authors described in the discussion, this analysis did not consider comorbidities. Given the older age of the "not high volume hospitals" patients, we assume they probably have many comorbidities. The distance to a high-volume hospital may make it impossible for them to go there, but the authors must not ignore this point.
3) The comparison was made by propensity score matching. The authors should state which method was used for PS matching.
Unbiased Propensity score matching requires at least weak confoundedness and overlap. The issue should be appropriately cleared up, especially if there is a clear causal link between the hospital and the comorbidity, but it is being ignored; it should be clearly stated.
4) The hub hospitals are defined for each region under the Basic Plan to Promote Cancer Control Program. The volume-outcome analysis should be done within the hub hospitals.
If the authors compare the top 25% of patient hospitals to others without considering the hospitals' facility status, the authors are not discussing the issue of volume alone. There is a misleading use of words.
Minor point
Since there is almost no data before 2005 for most cancers, there is likely a strong bias in the data up to 2005. The authors should comment on this.
Author Response
December 9, 2022
Dear Reviewer
Thank you for your review regarding our manuscript, “Volume-Outcome Relationship in Cancer Survival Rates: Analysis of a Regional Population-Based Cancer Registry in Japan ” which was assigned the manuscript number healthcare-2033848.
I have attached our revised manuscript and provided point-by-point responses to the reviewer’s valuable comments.
We wish to express our appreciation to the reviewer for their comments, which have helped us to significantly improve the paper.
Major point
- The authors define "High volume hospitals," but were these hospitals never changed in ranking over time?
Response: A review of changes in high-volume hospitals before and after 2005 showed that one hospital with a significant increase in beds was included as high volume for breast and bladder cancer from 2006 onwards. Otherwise, there was no change in the facilities included, although there was a reversal in the ranking within the high-volume bracket. This point has been added to the results. (lines 166-171)
2) As the authors described in the discussion, this analysis did not consider comorbidities. Given the older age of the "not high volume hospitals" patients, we assume they probably have many comorbidities. The distance to a high-volume hospital may make it impossible for them to go there, but the authors must not ignore this point.
Response: We have added your point to the limitations. (lines 312-318)
3) The comparison was made by propensity score matching. The authors should state which method was used for PS matching. Unbiased Propensity score matching requires at least weak confoundedness and overlap. The issue should be appropriately cleared up, especially if there is a clear causal link between the hospital and the comorbidity, but it is being ignored; it should be clearly stated.
Response: The PSM method is described in lines 120-126. The description of unmeasured confounding factors, which is a premise of the PSM, further emphasizes the possibility of a strong bias. Furthermore, a sensitivity analysis was performed on the results obtained by PSM. (lines126-127, 224-229、318-319)
4) The hub hospitals are defined for each region under the Basic Plan to Promote Cancer Control Program. The volume-outcome analysis should be done within the hub hospitals. If the authors compare the top 25% of patient hospitals to others without considering the hospitals' facility status, the authors are not discussing the issue of volume alone. There is a misleading use of words.
Response: Misleading and definitive statements have been revised. The wording has been changed from 'volume-outcome relationship is recognized' to 'high volume facilities may have a better prognosis', in some parts.
Minor point
Since there is almost no data before 2005 for most cancers, there is likely a strong bias in the data up to 2005. The authors should comment on this.
Response: Added to limitations for sample volume imbalance (line 320-322).
We feel that the revised manuscript reflects a suitable response to the comments and has been significantly improved over the initial submission. We trust that it is now suitable for publication in the Healthcare.
Sincerely.
Rena Kaneko M.D, Ph.D
Department of Gastroenterology, Japan Organization of Occupational Health and Safety Kanto Rosai Hospital.
1-1Kizukisumiyoshi-cho, Nakahara-ku, Kawasaki, Kanagawa, 211-8510 Japan
Tel: +81-44-411-3131; Fax: +81-44-411-3650
Reviewer 3 Report
The purpose of this study is to explore the relationship between hospital volume and outcomes with respect to cancer survival in Japan. Using a population-based cohort database from the Kanagawa cancer registry, the authors used propensity score matching to create a dataset for each of 10 common types of cancer. The authors used Cox proportional hazard modeling to calculate overall 5-year survival rate and hazard ratio (HR) for each type of cancer. Additional survival analyses were conducted for specific types of treatment (surgical, chemotherapy, etc.). The authors report that the HR was significantly lower among high volume hospitals for most common cancers except for bladder cancer. The HR ranged from .76 for stomach cancer to .85 for colon cancer. The authors conclude that cancer survival rates are in general better in high volume hospitals compared to other hospitals.
This is an interesting manuscript of high relevance to understanding the relationship of hospital volume to cancer survival rates in Japan. The analyses appear to be competently conducted and the results are important. Suggestions for strengthening the manuscript are listed below.
1. The format of the tables could be improved. At a minimum, Tables 1 and 2 should be formatted to be similar to Table 3. That is, the cancer type should be in bold font and there should be a blank line between the group of lines of data associated with each cancer condition. It would also help to reconfigure the tables to have indented text under each cancer type heading rather than centered text (that is, left justified test with indentation under each heading).
2. The tables are overwhelming and distract from the flow of the manuscript. Consider more effective ways to help the reader visualize the descriptive, HR, and Kapan-Meier results. It may even be possible to highlight the main findings in larger well-labelled figures and move some or all of the tables to an appendix. The figures are too small for the reader to decipher any of the text.
3. There are places where the writing is awkward and would benefit from some careful editing. For example, lines 248-257 on page 15 could be clearer.
Author Response
December 9, 2022
Dear Reviewer
Thank you for your review regarding our manuscript, “Volume-Outcome Relationship in Cancer Survival Rates: Analysis of a Regional Population-Based Cancer Registry in Japan ” which was assigned the manuscript number healthcare-2033848.
I have attached our revised manuscript and provided point-by-point responses to the reviewer’s valuable comments.
We wish to express our appreciation to the reviewer for their comments, which have helped us to significantly improve the paper.
- The format of the tables could be improved. At a minimum, Tables 1 and 2 should be formatted to be similar to Table 3. That is, the cancer type should be in bold font and there should be a blank line between the group of lines of data associated with each cancer condition. It would also help to reconfigure the tables to have indented text under each cancer type heading rather than centered text (that is, left justified test with indentation under each heading).
Response: Table 1 and table 2 have been corrected.
- The tables are overwhelming and distract from the flow of the manuscript. Consider more effective ways to help the reader visualize the descriptive, HR, and Kapan-Meier results. It may even be possible to highlight the main findings in larger well-labelled figures and move some or all of the tables to an appendix. The figures are too small for the reader to decipher any of the text.
Response: We considered moving them to the appendices, but as the comments from other reviewers suggested that there would be more tables, we have made the added tables as appendices and left Table1 and Table 2 in the main text. The figure values were made as large as possible and the whole figure was pasted in a larger size, but this was the limit of the amateur's editing skills. At the request of the editorial team, we are thinking of submitting the original figures as .pptx files and having them enlarged.
- There are places where the writing is awkward and would benefit from some careful editing. For example, lines 248-257 on page 15 could be clearer.
Response: Text that is difficult to understand, including lines 248-257 (now 271-277), has been revised.
We feel that the revised manuscript reflects a suitable response to the comments and has been significantly improved over the initial submission. We trust that it is now suitable for publication in the Healthcare.
Sincerely.
Rena Kaneko M.D, Ph.D
Department of Gastroenterology, Japan Organization of Occupational Health and Safety Kanto Rosai Hospital.
1-1Kizukisumiyoshi-cho, Nakahara-ku, Kawasaki, Kanagawa, 211-8510 Japan
Tel: +81-44-411-3131; Fax: +81-44-411-3650
Round 2
Reviewer 1 Report
Now the manuscript appears improved . Please consider just few other minor point regarding the following replies.
1) Response: In propensity score matching, the standard differences are unaffected by sample size. A standardized difference <0.1 indicates a negligible difference in the mean of proportion of a covariate between the case and control groups (Yasunaga H. Annals of Clinical Epidemiology 202;2(2)3:33-37, Park Y, et al. BMJ 2018 Mar 28:360:k1218). The literature also states, “showing p values is not recommended to check the balance of covariates, because failure to reject the null hypothesis does not guarantee successful balance of covariates between the groups”. Thus, we showed SD.
A note was added to the method on the points that indicated SD (lines 124-126). In fact, the tested P-values are significantly different for all cancers for both periods of diagnosis and stage. Please let me know if I still need to add more, or if my argument is incorrect.
I agree with the sentence “failure to reject the null hypothesis does not guarantee successful balance of covariates between the groups”; in fact, the rejection of null hypothesis says us that the distribution of the mean values difference (in case of continuous variables) of the two groups is ‘far enough’ from zero; of course this does not guarantee balance of covariates but this guarantees that there isn't a statistical difference in means after adjustment for covariates… anyway the Authors added explanation and references for making the reader aware and conscious enough for correctly interpreting the results…for me is ok.
2) Page 8, lines 188-189. This sentence appears to have no correspondence with the results reported in Table 3. With respect to 5-years survival rate, the better prognosis holds for: prostate , kidney and breast cancers, with 5-years survival rates (all cases) range from 74.4, 69.9 for kidney to 88.2 and 86.5 for breast.
Response: Sorry, lines 188 to 189 are correct as an explanation of the results in Table 3. Could you please elaborate on what is wrong? i.e., :“Esophagus, stomach, and lung cancer had a better prognosis for 5-year survival, and predominantly low HRs for high-volume hospital vs other volume for all eras and at all stages (adjusted HR ranged from 0.71-0.85 in esophagus, 0.69-0.82 in stomach and 0.78-0.90 in lung“. ( Now, lines 195-198).
Looking at the 5-years survival (all cases) the rates are:
|
|
High-volume |
Other-volume |
|
Esophagus |
36.8 |
30.1 |
|
stomach |
57.8 |
52.4 |
|
lung |
33.9 |
28.6 |
Thus the survival rate for these 3 cancers are lower with respect to , e.g., breast (88.2 and 86.5), colon (63.6, 58.8) and prostate (87.2, 83.7). So, I do not understand how the Authors can declare that the “Esophagus, stomach, and lung cancer had a better prognosis for 5-year survival”: I would be expecting a similar sentence for the 5-years survival for Breast or prostate . Please explain to avoid any misinterpretation.
Author Response
December 14, 2022
Dear Reviewer
Thank you for your review regarding our manuscript, “Volume-Outcome Relationship in Cancer Survival Rates: Analysis of a Regional Population-Based Cancer Registry in Japan ” which was assigned the manuscript number healthcare-2033848.
I have attached our revised manuscript and provided point-by-point responses to the reviewer’s valuable comments.
We wish to express our appreciation to the reviewer for their comments, which have helped us to significantly improve the paper.
2) Page 8, lines 188-189. This sentence appears to have no correspondence with the results reported in Table 3. With respect to 5-years survival rate, the better prognosis holds for: prostate , kidney and breast cancers, with 5-years survival rates (all cases) range from 74.4, 69.9 for kidney to 88.2 and 86.5 for breast.
Response: Sorry, lines 188 to 189 are correct as an explanation of the results in Table 3. Could you please elaborate on what is wrong? i.e., :“Esophagus, stomach, and lung cancer had a better prognosis for 5-year survival, and predominantly low HRs for high-volume hospital vs other volume for all eras and at all stages (adjusted HR ranged from 0.71-0.85 in esophagus, 0.69-0.82 in stomach and 0.78-0.90 in lung“. ( Now, lines 195-198).
Looking at the 5-years survival (all cases) the rates are:
|
|
High-volume |
Other-volume |
|
Esophagus |
36.8 |
30.1 |
|
stomach |
57.8 |
52.4 |
|
lung |
33.9 |
28.6 |
Thus the survival rate for these 3 cancers are lower with respect to , e.g., breast (88.2 and 86.5), colon (63.6, 58.8) and prostate (87.2, 83.7). So, I do not understand how the Authors can declare that the “Esophagus, stomach, and lung cancer had a better prognosis for 5-year survival”: I would be expecting a similar sentence for the 5-years survival for Breast or prostate . Please explain to avoid any misinterpretation.
Response:
Thank you for your detailed information. I understand it well now.The text was long, very confusing and redundant and has been corrected. We have also opened details of data on the prostate, kidneys and chest, as opposed to the descriptions on the esophagus, stomach and lungs. (lines 195-202)
We feel that the revised manuscript reflects a suitable response to the comments and has been significantly improved over the initial submission. We trust that it is now suitable for publication in the Healthcare.
Sincerely.
Rena Kaneko M.D, Ph.D
Department of Gastroenterology, Japan Organization of Occupational Health and Safety Kanto Rosai Hospital.
1-1Kizukisumiyoshi-cho, Nakahara-ku, Kawasaki, Kanagawa, 211-8510 Japan
Tel: +81-44-411-3131; Fax: +81-44-411-3

Reviewer 2 Report
I have read the revised document.
The authors used PSM, instead of IPW. The authors should state why they did not adopt IPW, or they should state whether there is a positivity violation.
I think that the authors used PSM to do KM analysis later. However, in that case, only matched cases can be compared. It should not be possible to conclude that the same is true for the entire population. It should be properly stated which population was being compared, in high-volume or others.
Author Response
December 14. 2022
Dear Reviewer
Thank you for your review regarding our manuscript, “Volume-Outcome Relationship in Cancer Survival Rates: Analysis of a Regional Population-Based Cancer Registry in Japan ” which was assigned the manuscript number healthcare-2033848.
I have attached our revised manuscript and provided point-by-point responses to the reviewer’s valuable comments. We wish to express our appreciation to the reviewer for their comments, which have helped us to significantly improve the paper.
The authors used PSM, instead of IPW. The authors should state why they did not adopt IPW, or they should state whether there is a positivity violation.
I think that the authors used PSM to do KM analysis later. However, in that case, only matched cases can be compared. It should not be possible to conclude that the same is true for the entire population. It should be properly stated which population was being compared, in high-volume or others.
Response:
Thank you for pointing this out.
We have added to the limitation that it does not faithfully reflect population effects. (lines318-322)
We feel that the revised manuscript reflects a suitable response to the comments and has been significantly improved over the initial submission. We trust that it is now suitable for publication in the Healthcare.
Sincerely.
Rena Kaneko M.D, Ph.D
Department of Gastroenterology, Japan Organization of Occupational Health and Safety Kanto Rosai Hospital.
1-1Kizukisumiyoshi-cho, Nakahara-ku, Kawasaki, Kanagawa, 211-8510 Japan
Tel: +81-44-411-3131; Fax: +81-44-411-3650
